# The Safety and Efficacy of Nusinersen in the Treatment of Spinal Muscular Atrophy: A Systematic Review and Meta-Analysis of Randomized Controlled Trials

**DOI:** 10.3390/medicina58020213

**Published:** 2022-02-01

**Authors:** Kirellos Said Abbas, Mennatullah Mohamed Eltaras, Nahla Ahmed El-Shahat, Basel Abdelazeem, Mahmoud Shaqfeh, James Robert Brašić

**Affiliations:** 1Faculty of Medicine, Alexandria University, Alexandria 22621, Egypt; kirellossaid98@gmail.com; 2Faculty of Medicine for Girls, Al-Azhar University, Cairo 11754, Egypt; mennatullah.mohamed20@gmail.com (M.M.E.); nahlaahmed752@gmail.com (N.A.E.-S.); 3Department of Medicine, McLaren Health Care, Flint, MI 48532, USA; baselelramly@gmail.com; 4Department of Medicine, Michigan State University, East Lansing 48824, MI, USA; 5Department of Neurology, McLaren Health Care, Flint, MI 48532, USA; mshaqfe1@gmail.com; 6Section of High Resolution Brain Positron Emission Tomography Imaging, The Russell H. Morgan Department of Radiology and Radiological Science, Division of Nuclear Medicine and Molecular Imaging, The Johns Hopkins University School of Medicine, Baltimore, MD 21287, USA

**Keywords:** adverse and beneficial effects, alpha motor neuron, bias, brainstem, evidence, neurodegenerative, prevalence, risk, spinal cord, survival of motor neuron 1 (*SMN1*) gene

## Abstract

*Background and objectives*: Spinal muscular atrophy (SMA) is a neurodegenerative disease that leads to progressive proximal muscle weakness and muscle atrophy. To assess the beneficial and adverse effects of nusinersen, a promising intervention for SMA, we conducted a systematic search and meta-analysis of the published randomized control trials (RCTs) of nusinersen for SMA. *Materials and methods*: Utilizing the Preferred Reporting for Systematic Review and Meta-Analysis (PRISMA), we searched PubMed, Scopus, Web of Science, Cochrane Central, and Clinicaltrials.gov from inception to 22 July 2021. *Results*: Three RCTs satisfying the inclusion and exclusion criteria covered 274 patients: 178 patients in the nusinersen group. Our results show a significant risk difference (RD) in the motor milestone response (RD: 0.51; 95% CI: 0.39, 0.62; *p* < 0.00001) and improvement in the HINE-2 score (RD: 0.26; 95% CI: 0.12, 0.40; *p* < 0.0003) in the nusinersen group compared to the control group. Moreover, a significant decrease in the risk ratio (RR) for severe adverse events (RR: 0.72; 95% CI: 0.57, 0.92; *p* = 0.007) and any adverse event leading to treatment discontinuation (RR: 0.40; 95% CI: 0.22, 0.74; *p* = 0.004) was observed. An insignificant result was found for any adverse effects (RR: 0.93; 95% CI: 0.97, 1.01; *p* = 0.14) and for serious adverse effects (RR: 0.81; 95% CI: 0.60, 1.07; *p* = 0.14). *Conclusions*: This review provides evidence that nusinersen treatment was effective in treatment for infants with SMA and was associated with fewer severe adverse events; however, more RCTs are needed to establish evidence.

## 1. Introduction

### 1.1. Background

Spinal muscular atrophy (SMA) is a neurodegenerative disease characterized by the degeneration of the alpha motor neurons in the spinal cord and motor nuclei in the lower brainstem leading to progressive proximal muscle weakness and muscle atrophy [1]. Without treatment, SMA is a leading genetic cause of infant death, with an estimated prevalence of 8.5–10.3 per 100,000 live births [2]. With ventilatory support and other treatments, milestones are attained and survival increases up to 70% [3,4,5].

SMA can be classified into five clinical grades, depending on the age of onset and the severity of the disease [6]. SMA, which is an autosomal recessive disorder, is caused by the loss through deletions or mutations of the survival of motor neuron 1 (*SMN1*) gene on chromosome 5q13.2, leading to a deficiency of the survival of motor neuron (SMN) protein [7,8], required for the maintenance of motor neurons [9].

Chromosome 5q13.2 contains both the telomeric *SMN1* gene and the centromeric survival of the motor neuron 2 (*SMN2*) gene [10]. The main difference between the *SMN1* and *SMN2* genes is the transition from cytosine to thymidine in exon 7 in the *SMN2* gene, leading to the deficiency of exon 7 and the production of a truncated and non-functional SMN protein. However, approximately 10 to 15 percent of *SMN2* messenger ribonucleic acid (mRNA) contains exon 7 and produces some full-length, functional SMN proteins [6]. Thus, the loss of the SMN protein resulting from the *SNM1* gene is partially compensated for by the synthesis of the SMN protein resulting from the *SNM2* gene.

### 1.2. A Novel Intervention to Increase Full-Length, Functional SMN Proteins

Nusinersen is an antisense oligonucleotide drug that increases the production of the SMN protein by modifying the pre-messenger ribonucleic acid (pre-mRNA) splicing of the SMN protein. Nusinersen acts by binding to a repressive splicing sequence in the intron 7, leading to the inclusion of exon 7 in the pre-mRNA transcript. As a result, full-length, functional SMN proteins are increased [11].

In the last few years, there has been a growing interest in nusinersen. The EMBRACE RCT concluded that there were no nusinersen-related adverse events (AEs) after following a patient for 14 months and reported improvement in motor function with nusinersen after 24 months [12]. On the other hand, the ENDEAR RCT showed improvement in motor functions, supporting the early usage of nusinersen in SMA [13]. However, since no meta-analysis on nusinersen had been conducted yet, we conducted a systematic review and meta-analysis aiming to evaluate the safety and efficacy of nusinersen in treating SMA patients to provide reliable data to clinicians caring for patients with SMA.

## 2. Materials and Methods

### 2.1. Data Sources and Search Strategy

On 22 July 2021, K.S.A. and M.M.E. searched PubMed, Scopus, Web of Science, and Cochrane Central for possible included RCTs using the following search term: (CDR132l or nusinersen or IONIS-SMNRX or ISIS-SMNRX) AND (spinal muscular atrophy or SMA). A further search was performed manually through Clinicaltrials.gov and related articles [14]. The search results were exported to EndNote [15]. The duplicates were removed, and one file was exported in Microsoft Excel format to start screening. Table 1 shows the overall search method.

The Preferred Reporting for Systematic Review and Meta-Analysis (PRISMA) [16] checklist items for conducting, reporting, and writing with the corresponding pages in this article are reported in Appendix A and Appendix A.

On 31 August 2021, the protocol was registered in the PROSPERO International prospective register of systematic reviews with the registration number CRD42021270037 (https://www.crd.york.ac.uk/prospero/display_record.php?RecordID=270037) (accessed on 27 November 2022).

### 2.2. Eligibility Criteria

#### 2.2.1. Inclusion Criteria

Only RCTs in the English language discussing the safety and efficacy of CDR132l or nusinersen for patients with SMA were included in the meta-analysis.

#### 2.2.2. Exclusion Criteria

Study designs other than RCTs were excluded. Languages other than English were excluded.

### 2.3. Study Selection

Screening for titles/abstracts and full texts was conducted independently by K.S.A. and M.M.E. Any conflicts between K.S.A. and M.M.E. were resolved through a consensus conference with B.A.

### 2.4. Data Extraction

N.A.E.-S. and M.M.E. extracted the data: the baseline characteristics, a summary of the included studies, and the outcomes. Any conflict in opinion between N.A.E.-S. and M.M.E. was resolved in a consensus conference with K.S.A. The data for the summary and baseline were extracted, including the author name, year of publication, drug of intervention, control group, number in the intervention group, number in the control group, gender, and follow-up duration. Outcomes related to milestones and adverse effects were also extracted. The final tables were designed and arranged for optimal presentation.

### 2.5. Risk-of-Bias Assessment

In order to identify trustworthy evidence [17,18], N.A.E.-S. and M.M.E. used The Cochrane Collaboration’s tool for assessing the risk of bias in randomized trials to evaluate selection, performance, detection, attrition, reporting biases, and any other bias [19]. The overall grade of each aspect was measured as a low risk, high risk, or unclear risk of bias. Conflicts between the authors were resolved through conferences to establish a consensus.

### 2.6. Outcome of Interest

The primary outcome of interest was the motor milestone response, based on the Hammersmith Infant Neurological Examination (HINE-2) score. The HINE-2 score had a standardized assessment tool for motor milestones, including head control, sitting, voluntary grasp, ability to kick in supine, rolling, crawling or bottom shuffling, standing, and walking. The scoring system ranged from 0 to 26, with a higher score representing a better neurological performance [20]. The outcomes for the adverse events were also analyzed, including any adverse event, any adverse event leading to treatment discontinuation, any severe adverse event, and any serious adverse event.

### 2.7. Statistical Analysis

RevMan manager v5.3 (Cochrane, London, UK) was used by N.A.E.-S. and K.S.A. to analyze the extracted data [21]. Pooled risk ratios (R.R.s) or risk differences (R.D.s) were used for dichotomous data. The results are represented with 95% confidence intervals (CIs) determined through the Mantel–Haenszel method. The use of the random-effects model or fixed-effects model depended on the statistical value of the heterogeneity in each outcome. If there was significant heterogeneity I^2^ > 50, a random-effects model was used, and if I^2^ < 50, a fixed-effects model was used [22]. Due to the small number of included studies, the publication bias was not assessed [23].

## 3. Results

### 3.1. Search Results and Study Selection

The systematic search revealed 1262 articles. We removed the 472 items identified as duplicates by EndNote. Finally, 790 articles were screened. Figure 1 represents the PRISMA flowchart, including the details of the whole screening process with the mentioned reasons for exclusion [16]. Three articles met the criteria for our study [12,13,24]. No additional articles were included manually.

### 3.2. Characteristics of Included Studies

The three included RCTs were double-blinded. Two RCTs, the ENDEAR [13] and CHERISH [24] trials, were phase 3 and global, and the third one, the EMBRACE trial [12], was phase 2 and performed in the United States of America (USA) and Germany [12]. The total number of participants was 274, including 178 patients in the nusinersen group and 96 patients in the control group. Females represented 51.8% of the total participants. One phase 3 study, the ENDEAR trial [13], included only infantile-onset SMA, and the other phase 3 study, the CHERISH trial [24], included only later-onset SMA [24]. The phase 2 study, the EMBRACE trial [12], included both types [12]. The median age at symptom onset ranged from 5.3 to 33.6 months, with the median age of SMA diagnosis ranging from 10.6 to 56.8 months. The characteristics of the included studies and the patients’ baseline characteristics are summarized in Table 2.

### 3.3. Risk of Bias (ROB)

The risks of biases for each study are indicated in Figure 2.

### 3.4. Outcomes of Interest

#### 3.4.1. Primary Outcomes of Interest

The two RCTs with data for an analysis of the motor milestone response and HINE-2 score [12,13] showed a significant risk difference (RD) in the motor milestone response (RD: 0.51; 95% CI: 0.39, 0.62; *p* < 0.00001) and improvement in HINE-2 score (RD: 0.26; 95% CI: 0.12, 0.40; *p* < 0.0003) in the nusinersen group compared to control group (Figure 3).

#### 3.4.2. Secondary Outcomes of Interest

All three RCTs [12,13,24] reported the outcomes of adverse events. There were significant decreases in severe adverse events (RR: 0.72; 95% CI: 0.57, 0.92; *p* = 0.007) and in any adverse event leading to treatment discontinuation (RR: 0.40; 95% CI: 0.22, 0.74; *p* = 0.004). An insignificant result was found for any adverse effects (RR: 0.93; 95% CI: 0.97, 1.01; *p* = 0.14) and serious adverse effects (RR: 0.81; 95% CI: 0.60, 1.07; *p* = 0.14) (Figure 4). The most common adverse effects included pyrexia, vomiting, constipation, cough, upper-respiratory-tract infection, and pneumonia. Table 3 shows the numbers of patients with common adverse effects in both the nusinersen group and control group.

## 4. Discussion

### 4.1. Safety and Efficacy of Nusinesen in SMA

To our knowledge, this is the first study to meta-analyze the safety and efficacy of nusinersen in SMA and to determine the significance of the results and the impact of the new data on clinical practice. Our results show that the infants in the nusinersen group achieved clinical meaningful motor responses and overall improvement in neuromuscular function with nusinersen compared to the control group. Despite nusinersen’s promising efficacy results, none of the babies administered nusinersen achieved normal motor development, some required continuous feeding and mechanical ventilation, and some babies even died. These results show that nusinersen is not curative in symptomatic patients. However, our results showed improvement in total milestones. The motor milestone was a term used to describe the overall physical and development parameters in the affected patients. Mercuri et al. [24] and Acsadi et al. [12] detailed the improvement in milestones through the ability to sit with/without the support and the ability to walk more than 15 feet independently. Finkel et al. [13] added head control, rolling, crawling, kicking, and voluntary grasp.

According to Coratti et al. [25], age was a significant predictive parameter for a positive response for the treatment depending on the Hammersmith Functional Motor Scale Expanded (HFMSE) scoring system in a sample of type II SMA. De Vivo et al. [26] recommended pre-symptomatic treatment for genetically detected patients: SMA I and II. By contrast, Hagenacker et al. [27] postulated no role of age in treatment in 5q SMA and attributed the variable responses to the subtypes of SMA gene defect.

### 4.2. Adverse Effects

Nusinersen is administered intrathecally to ensure delivery to the central nervous system (CNS) due to its poor ability to cross the blood–brain barrier (BBB), with an estimated half-life of 135 to 177 days [28]. Lumbar puncture (LP) for the intrathecal administration of nusinersen carries some challenges in patients with scoliosis, a manifestation of SMA, so alternative routes must be considered such as subarachnoid and cervical punctures [28,29,30]. LP brings multiple possible complications including post-LP headache, vomiting, back pain, bleeding, and infection. Of these, vomiting was the most common that occurred within 72 h of the LP in the EMBRACE trial [12]. The most common AEs noted in the nusinersen group were pyrexia and upper-respiratory-tract infections [12,13,24] (Table 3).

Constipation was a common AE in the nusinersen group [13]. There were no study discontinuations due to nusinersen-related AEs in the EMBRACE and ENDEAR trials [12,13]. Nevertheless, the ENDEAR trial reported that the numbers of events leading to the discontinuation of the study drug were 13 of 80 in the nusinersen group, in contrast to 16 of 41 in the control group [13].

Finally, respiratory distress was higher in the nusinersen group in [12,13,24]. Still, the overall mean percentage time on ventilator support (including bilevel positive airway pressure, intubation, tracheostomy, and endotracheal tubes) was 11.3%, lower than that in the control group, 29.8% [12].

Our analysis revealed that the nusinersen group had fewer severe adverse events and any adverse events leading to treatment discontinuation compared to the control group; however, both groups were similar in terms of any adverse effects and serious adverse effects.

### 4.3. Cost of Nusinersen

The high cost of the nusinersen, even for high-income countries, makes ensuring cost-effectiveness essential for supporting the drug’s usage in clinical discussions [31]. A simulation in Sweden of the cost of effectiveness of nusinersen using a Markov cohort model of the ENDEAR [13] and CHERISH trials [24] identified that the incremental cost of nusinersen exceeded EUR 2 million and that nusinersen was not cost-effective when using the willingness-to-pay threshold of EUR 195,600 [32]. A similar study in the USA showed that the incremental cost-effectiveness ratio of nusinersen with screening was USD 330,558 per event-free life year (LY) saved, compared to USD 508,481 for nusinersen treatment without screening [33]. In order for nusinersen to be cost-effective with a willingness-to-pay threshold of USD 50,000 per event-free LY saved, the dose price should be USD 23,361 instead of the current price of USD 125,000 [33].

### 4.4. Limitations

The trustworthiness of the evidence [18] in meta-analyses is enhanced by the inclusion of multiple comparable studies with large samples in multiple locations throughout the world [34,35]. Instead of merely conducting a selective review of extant studies [36], we chose to utilize the state-of-the-art tools for an optimal systematic review and meta-analysis.

Our meta-analysis was limited by the discrepancies in the study designs of the selected RCTs. Although the ENDEAR trial of only participants with infantile-onset SMA [13] used both the HINE-2 score [20] and the Children’s Hospital of Philadelphia Infant Test of Neuromuscular Disorders (CHOP INTEND) score [37,38] to measure the motor milestone response, the EMBRACE trial of participants with both infantile-onset and later-onset SMA [12] and the other phase 3 trial, the CHERISH trial, of only participants with later-onset SMA [24] used only the HINE-2 score [20]. The CHOP INTEND score is specific for patients with SMA [37,38], in contrast to the HINE-2 score, which was used to score clinical neurological examinations for infants between 2 and 24 months of age, but it could be used to assess infants with SMA [39]. Future investigations will be enhanced by the use of identical inclusion criteria and outcome measures. More stringent criteria for the selection of articles, including ages and rating scales, will enhance future reviews when there are many more articles to include.

A further limitation of the present meta-analysis is the small number of available RCTs with outcomes that were suitable for pooling through meta-analysis; thus, more RCTs are needed before making clinical recommendations based on these studies. Currently, one more RCT is in the recruiting stage and assessing the same outcomes (NCT04089566).

## 5. Conclusions

This review provides evidence that nusinersen treatment was effective in the treatment of infants with SMA and was associated with fewer severe adverse events when compared to the control group. Additional well-designed RCTs with identical inclusion and exclusion criteria and assessment measures and longer follow-up periods by multiple investigators in diverse locations are needed before a definitive systematic review and meta-analysis can be conducted. However, the initial results for the safety and efficacy of nusinersen are promising.

## Figures and Tables

**Figure 1 medicina-58-00213-f001:**
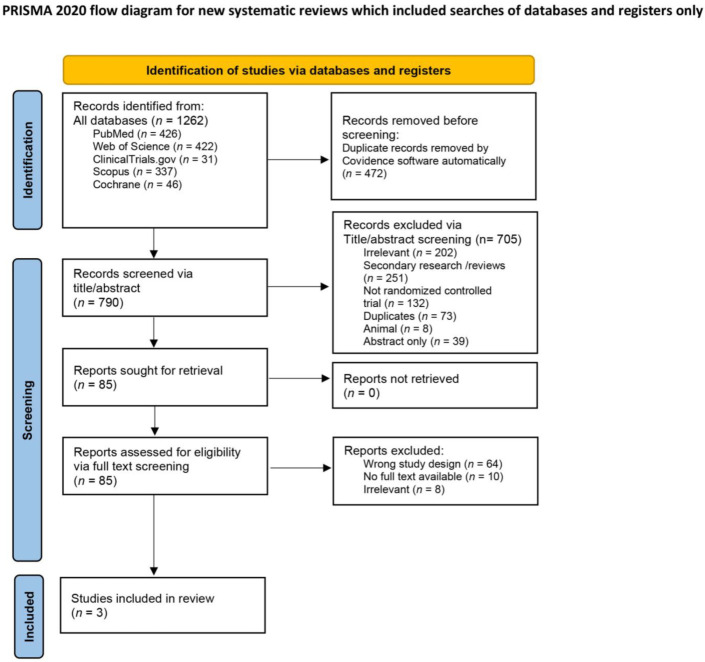
PRISMA 2020 flow diagram for updated systematic reviews [16].

**Figure 2 medicina-58-00213-f002:**
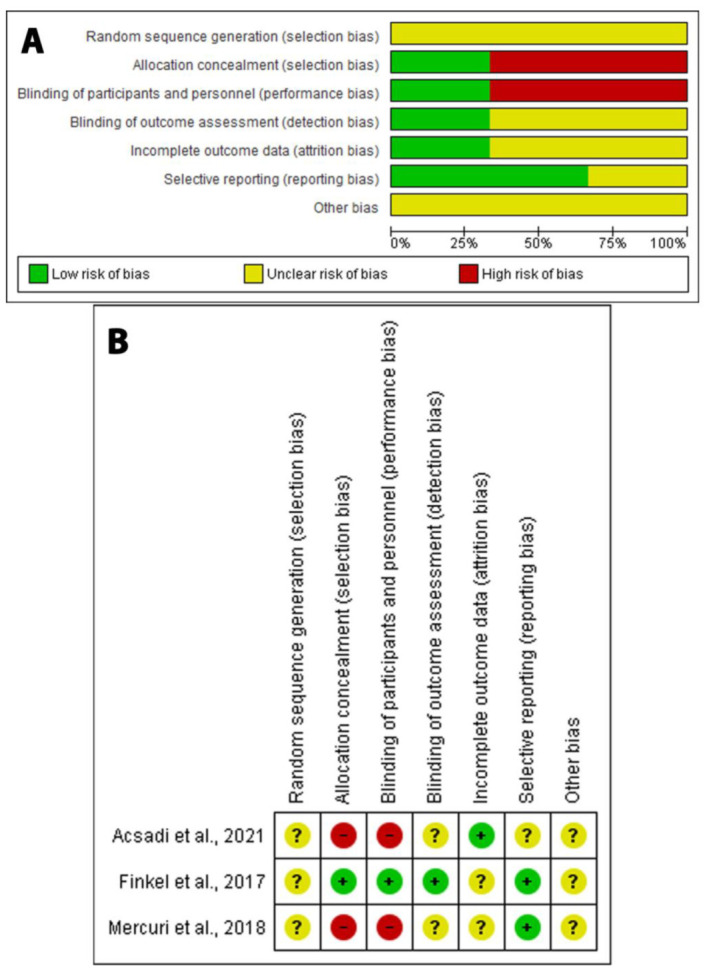
Quality assessment of the risks of biases in the studies in the meta-analysis. (**A**) The upper panel represents risks (low, unclear, and high) for the subtypes of biases of the combination of studies included in this review. (**B**) The lower panel presents a schematic representation of risks (low = red, unclear = yellow, and high = red) for specific types of biases of each of the studies in the review.

**Figure 3 medicina-58-00213-f003:**
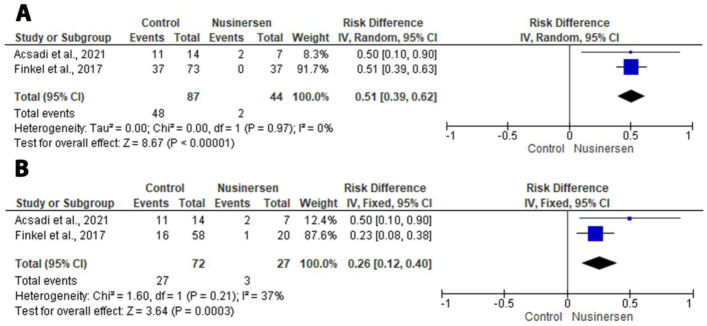
Forest plot of the primary outcomes. (**A**) Forest plot of the motor milestone response. (**B**) Forest plot of the improvement in HINE-2 score [20].

**Figure 4 medicina-58-00213-f004:**
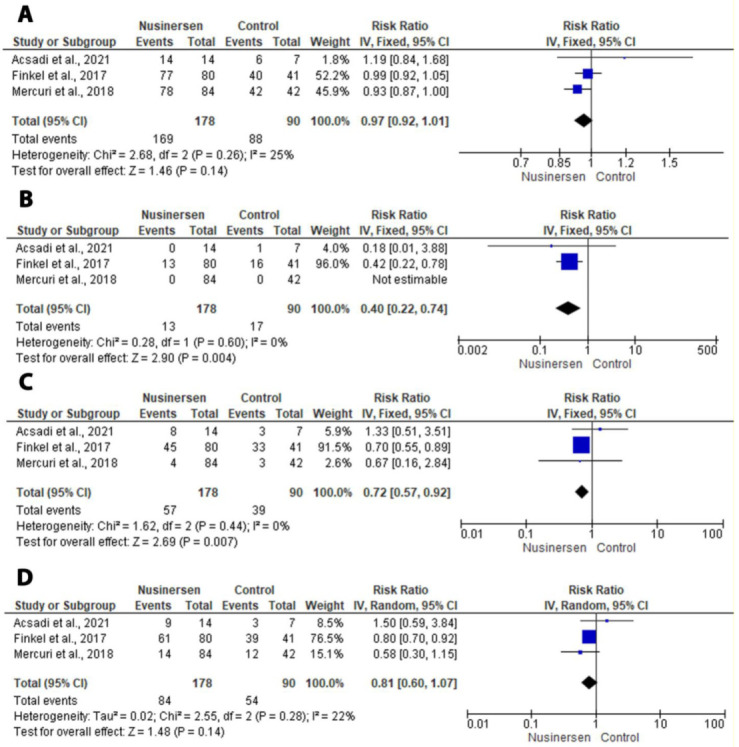
Forest plot of the secondary outcomes. (**A**) Forest plot of any adverse event. (**B**) Forest plot of any adverse event led to treatment discontinuation. (**C**) Forest plot of severe adverse event. (**D**) Forest plot of serious adverse event.

**Table 1 medicina-58-00213-t001:** Search terms in databases.

Database Name	Search Terms	Results
Scopus	(CDR132l or nusinersen or IONIS-SMNRX or ISIS-SMNRX) and (spinal muscular atrophy or SMA)	337
PubMed	(CDR132l or nusinersen or IONIS-SMNRX or ISIS-SMNRX) and (spinal muscular atrophy or SMA)	426
Web of science	(CDR132l or nusinersen or IONIS-SMNRX or ISIS-SMNRX) and (spinal muscular atrophy or SMA)	442
Cochrane	(CDR132l or nusinersen or IONIS-SMNRX or ISIS-SMNRX) and (spinal muscular atrophy or SMA)	46
Clinicaltrials.gov	CDR132l or nusinersen or IONIS-SMNRX or ISIS-SMNRX	31

**Table 2 medicina-58-00213-t002:** Characteristics of the studies selected for the meta-analysis.

Author, Year	Country	Study Design	Population	Study Phase	Total Sample Size	Treatment (N)	Control (N)	Female Sex Treated by Nusinersen, N (%)	Type of SMA (N)	Median Age at Symptom Onset, Months	Median Age at Diagnosis, Months	Observations
Acsadi et al., 2021 (EMBRACE trial) [12]	USA and Germany	Double-blind RCT	Infants and young children	2	21	Nusinersen (14)	Sham in Part 1 (7)/ nusinersen in Part 2 (6)	5 (36)	Infantile-onset (13)/ later-onset (8)	5.3	10.6	Both infantile and late-onset SMA showed a long-term benefit–risk ratio. No drug-related adverse effects or discontinuation due to the drug. Milestone improvement was 93% in the treatment group vs. 29% in the sham group.
Finkel et al., 2017 (ENDEAR trial) [13]	31 global centers	Double-blind RCT	Infants	3	121	Nusinersen (80)	Sham (41)	43 (54)	Infantile onset (121)	2.1 (mean)	3.5 (mean)	The treatment group had a higher probability of living longer. The hazard ratio for death was 0.53 vs. 0.37 in the sham group. Early treatment may be more beneficial for milestone improvement.
Mercuri et al., 2018 (CHERISH trial) [24]	24 global centers	Double-blind RCT	Children	3	126	Nusinersen (84)	Sham (42)	46 (55)	Later-onset (126)	10.3	18	There was a significant improvement regarding milestones in late-onset SMA. Adverse effects were similar in both groups, but there was a higher improvement in the HFMSE score in the treatment group vs. sham.

RCT: Randomized control trial; SMA: Spinal muscular atrophy; USA: United States of America; HFMSE: Hammersmith Functional Motor Scale—Expanded [20].

**Table 3 medicina-58-00213-t003:** Common adverse events.

Author, Year	Treatment (*N*)	Pyrexia	Vomiting	Constipation	Cough	Upper-Respiratory-Tract Infection	Pneumonia
	Control (*N*)						
Acsadi et al., 2021 (EMBRACE trial) [12]	Nusinersen (14)	12	8	4	11	6	9
	Sham (7)	1	1	1	1	1	0
Finkel et al., 2017 (ENDEAR trial) [13]	Nusinersen (80)	45	14	28	9	24	23
	Sham (41)	24	8	9	8	9	7
Mercuri et al., 2018 (CHERISH trial) [24]	Nusinersen (84)	36	24	1	21	25	2
	Sham (42)	15	5	0	9	19	6

*N*: Number of patients.

## Data Availability

All the data included in this report are identified within the text, tables, and figures.

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
