# Peer review of "The Safety and Efficacy of Nusinersen in the Treatment of Spinal Muscular Atrophy: A Systematic Review and Meta-Analysis of Randomized Controlled Trials"

_medicina, 2022, doi:10.3390/medicina58020213_

Round 1
Reviewer 1 Report
In this review article, Abbas K S et al has done a comprehensive search of the articles, published regarding the case studies of beneficial and adverse effects of Nusinersen, an antisense oligo drug for SMA therapy. Authors went through a systematic search process of the published works in 5 different databases and found 1262 papers using some specific key words. Thereafter authors used Systematic reviews and Meta-Analyses (PRISMA) by setting up some inclusion and exclusion criteria to find out the suitable and well-designed studies for the meta-analysis. Finally came up with three studies, which were included in the analysis of beneficial and adverse effects of Nusinersen in SMA.
Nusinersen is the only FDA approved anti sense oligo drug for the treatment of SMA. It has some side effects also. In this context this review does not give any new information except the study design for using PRISMA to identify the articles showing well systematic studies for the Nusinersen treatment.
Higher stringency for inclusion/ exclusion criteria might remove some of the important articles, which may add more importance in this present review. There are 5 types of MSA categorized according to the age of onset of disease. Discussion of differential effect of Nusinersen treatment in the different age group can add more weightage in this review.
Overall, this review is well written and shows a way to make a systematic review process and meta-analysis from the published articles from the different databases.
Author Response
Point 1: I In this review article, Abbas K S et al has done a comprehensive search of the articles, published regarding the case studies of beneficial and adverse effects of Nusinersen, an antisense oligo drug for SMA therapy. Authors went through a systematic search process of the published works in 5 different databases and found 1262 papers using some specific key words. Thereafter authors used Systematic reviews and Meta-Analyses (PRISMA) by setting up some inclusion and exclusion criteria to find out the suitable and well-designed studies for the meta-analysis. Finally came up with three studies, which were included in the analysis of beneficial and adverse effects of Nusinersen in SMA.
Nusinersen is the only FDA approved anti sense oligo drug for the treatment of SMA. It has some side effects also. In this context this review does not give any new information except the study design for using PRISMA to identify the articles showing well systematic studies for the Nusinersen treatment.
Higher stringency for inclusion/ exclusion criteria might remove some of the important articles, which may add more importance in this present review.
Response 1:
Thank you so much for your thoughtful comment and valuable suggestions to improve our manuscript. your care of high-quality research.
To ensure high-quality research, we restricted our criteria to randomized control trials, which provide well-established design, blinding, and controlling. The utilization of more stringent criteria, such as the ages of the participants and the utilization of specific rating scales, will be included in future reviews when there are many more articles. However, the application of such stringent criteria now would lead to single articles. We addressed this issue in the last sentence of the second-to-last paragraph of 4.4 Limitations as follows:
More stringent criteria for selection of articles, including ages and rating scales, will enhance future reviews when there are many more articles to include.
Point 2: There are 5 types of MSA categorized according to the age of onset of disease. Discussion of differential effect of Nusinersen treatment in the different age group can add more weightage in this review.
Response 2:
Thank you so much for your insightful comment. To address this issue, we added the following paragraph at the end of 4.1. Safety and efficacy of nusinesen in SMA
According to Coratti et al. [25], age was a significant predictive parameter for a positive response for the treatment depending on the Hammersmith Functional Motor Scale Expanded (HFMSE) scoring system in a sample of type II SMA. De Vivo et al. [26] recommended pre-symptomatic treatment for genetically detected patients; SMA I and II. In contrast, Hagenacker et al. [27] postulated no role of age in treatment in 5q SMAand attributed the variable responses to the subtypes of SMA gene defect.

Reviewer 2 Report
In this paper titled “The Safety and Efficacy of Nusinersen in the Treatment of Spinal Muscular Atrophy: A Systematic Review and Meta-analysis of Randomized Controlled Trials” The author reviewed and performed met-analysis of randomized controlled trails to study safety and efficacy of Nusinersen for the treatment of Spinal muscular atrophy. The research in this article is original and well-designed and executed. However, data presentation may be improved (Figure 1 shows only results with no explanation for why an article was omitted or included), and the findings were not described clearly enough to support the assertion that nusinersen treatment was effective in the treatment of infants with SMA.
Author Response
Point 1: I In this paper titled “The Safety and Efficacy of Nusinersen in the Treatment of Spinal Muscular Atrophy: A Systematic Review and Meta-analysis of Randomized Controlled Trials” The author reviewed and performed met-analysis of randomized controlled trails to study safety and efficacy of Nusinersen for the treatment of Spinal muscular atrophy. The research in this article is original and well-designed and executed. However, data presentation may be improved (Figure 1 shows only results with no explanation for why an article was omitted or included),
Response 1:
Thank you so much for your valuable comment. We revised figure 1 according to your comment as follows:
Figure 1. PRISMA 2020 flow diagram for updated systematic reviews [16].
Point 2: and the findings were not described clearly enough to support the assertion that nusinersen treatment was effective in the treatment of infants with SMA.
Response 2:
Thank you so much for your thoughtful comment. To address this issue, we added a discussion at the end of the first paragraph of 4.1. Safety and efficacy of nusinesen in SMA as follows:
However, our results found improvement in total milestones. The motor milestone was a term used to describe the overall physical and development parameters in the affected patients. Mercuri et al. [24] and Acsadi et al. [12] detailed the improvement in milestones through the ability to sit with/without the support and the ability to walk more than 15 feet independently. Finkel et al. [13] added head control, rolling, crawling, kicking, and voluntary grasp.

Reviewer 3 Report
This is article covers a study in a topic which presents a subtype of unmet medical need. Neurodegenerative hereditary conditions remain highly untreated despite the efforts of the scientific and medical community in the past decades. So any effort towards identifying potential agents to mitigate the effects and allow for patients to have at least partially controlled condition is worth a try.
Introduction provides clear and concise background of the condition and the studied molecule.
There is a repetition on Lines 74-78 repeating the text and references of lines 40-43
Materials and methods describe in a very detailed manner the process that was utilized although the section contains detailed description on which author did what. This repeats the information provided as post script to the article.
Results however need further elaboration. It is mentioned that the primary outcome HINE-2 is significantly improved in all nusinersen groups is insufficient as a result of a meta-analysis of this scale. It would be useful authors to include a table with selection of efficiency outcomes and observations of all reviewed studies.
The secondary outcome – safety is also discussed in a very high level manner. The best way to visualize the good safety profile is through a table with listed all major adverse events in nusinersen and in control group.
Discussion section needs to be further developed too.
Author Response
Point 1: Introduction provides clear and concise background of the condition and the studied molecule.
There is a repetition on Lines 74-78 repeating the text and references of lines 40-43
Response 1:
Thank you so much for your thoughtful comments. We appreciate your feedback and your efforts to improve our manuscript. We removed the repetition on Lines 74-78 and the references of lines 40-43.
Point 2: Materials and methods describe in a very detailed manner the process that was utilized although the section contains detailed description on which author did what. This repeats the information provided as post script to the article.
Response 2:
Thank you so much for your thoughtful comment. Following the PRISMA checklist, we mentioned how many authors and which author did what in the manuscript itself. In order to provide a crystal clear explanation of the process to readers who may be unfamiliar with the PRISMA format, we also describe the process in the text.
Point 3: Results however need further elaboration. It is mentioned that the primary outcome HINE-2 is significantly improved in all nusinersen groups is insufficient as a result of a meta-analysis of this scale. It would be useful authors to include a table with selection of efficiency outcomes and observations of all reviewed studies.
Response 3:
Thank you so much for your thoughtful comment. We added the requested information to Table 2. Thanks for helping us to improve our article.
Point 4: The secondary outcome – safety is also discussed in a very high level manner. The best way to visualize the good safety profile is through a table with listed all major adverse events in nusinersen and in control group.
Response 4:
Thank you so much for your comment. Thanks for helping us to improve our manuscript. To address this issue, we added a narrative and a table at the end of 3.4.2. Secondary outcomes of interest as follows:
The most common adverse effects included pyrexia, vomiting, constipation, cough, upper respiratory tract infection, and pneumonia. Table 3 shows the numbers of patients with common adverse effects in both nusinersen group and control group.
Table 3. Common adverse events.
|
Author, year |
Treatment (N) |
Pyrexia |
Vomiting |
Constipation |
Cough |
Upper respiratory tract infection |
Pneumonia |
|
Control (N) |
|||||||
|
Acsadi, et al., 2021 (EMBRACE trial) [12] |
Nusinersen (14) |
12 |
8 |
4 |
11 |
6 |
9 |
|
Sham (7) |
1 |
1 |
1 |
1 |
1 |
0 |
|
|
Finkel, et al., 2017 (ENDEAR trial) [13] |
Nusinersen (80) |
45 |
14 |
28 |
9 |
24 |
23 |
|
Sham (41) |
24 |
8 |
9 |
8 |
9 |
7 |
|
|
Mercuri, et al., 2018 (CHERISH trial) [24] |
Nusinersen (84) |
36 |
24 |
1 |
21 |
25 |
2 |
|
Sham (42) |
15 |
5 |
0 |
9 |
19 |
6 |
|
|
N: Number of patients |
|
|
|
|
|
|
|
Point 5: Discussion section needs to be further developed too.
Response 5:
Thank you so much for the thoughtful comment. We added more paragraphs to develop our discussion at the end of 4.1. Safety and efficacy of nusinesen in SMA as follows:
. However, our results found improvement in total milestones. The motor milestone was a term used to describe the overall physical and development parameters in the affected patients. Mercuri et al. [24] and Acsadi et al. [12] detailed the improvement in milestones through the ability to sit with/without the support and the ability to walk more than 15 feet independently. Finkel et al. [13] added head control, rolling, crawling, kicking, and voluntary grasp.
According to Coratti et al. [25], age was a significant predictive parameter for a positive response for the treatment depending on the Hammersmith Functional Motor Scale Expanded (HFMSE) scoring system in a sample of type II SMA. De Vivo et al. [26] recommended pre-symptomatic treatment for genetically detected patients; SMA I and II. In contrast, Hagenacker et al. [27] postulated no role of age in treatment in 5q SMA and attributed the variable responses to the subtypes of SMA gene defect.

Round 2
Reviewer 2 Report
The author reviewed and performed a meta-analysis of randomized controlled trials to study the safety and efficacy of Nusinersen for the treatment of Spinal muscular atrophy in this paper titled "The Safety and Efficacy of Nusinersen in the Treatment of Spinal Muscular Atrophy: A Systematic Review and Meta-analysis of Randomized Controlled Trials." This article's research is unique, well-designed, and implemented.
This manuscript is a resubmission of an earlier submission. The following is a list of the peer review reports and author responses from that submission.